# Effectiveness of anti-vascular endothelial growth factors in neovascular age-related macular degeneration and variables associated with visual acuity outcomes: Results from the EAGLE study

**Giovanni Staurenghi**[1]*, **Francesco Bandello**[2], **Francesco Viola**[3], **Monica Varano**[4], **Giulia Barbati**[5], **Elena Peruzzi**[6], **Stefania Bassanini**[6], **Chiara Biancotto**[6], **Vito Fenicia**[7], **Claudio Furino**[8], **Maria Vadalà**[9], **Michele Reibaldi**[10], **Stela Vujosevic**[11], **Federico Ricci**[12], on behalf of the EAGLE study investigators[¶]

1 Department of Biomedical and Clinical Sciences "Luigi Sacco", Luigi Sacco Hospital, University of Milan, Milan, Italy, 2 Department of Ophthalmology, IRCCS Ospedale San Raffaele, University Vita-Salute, Milan, Italy, 3 Foundation IRCCS Cà Granda Ospedale Maggiore Policlinico, University of Milan, Milan, Italy, 4 IRCCS-Fondazione Bietti, Rome, Italy, 5 Department of Medical Sciences, Biostatistics Unit, University of Trieste, Trieste, Italy, 6 Novartis Farma S.p.A., Origgio, VA, Italy, 7 Faculty of Medicine and Psychology, NESMOS Department, Ophthalmology Unit, S. Andrea Hospital, University of Rome "La Sapienza", Rome, Italy, 8 Department of Medical Science, Neuroscience and Sense Organs, Eye Clinic, Azienda Ospedaliero-Universitaria Policlinico Consorziale Bari, Bari, Italy, 9 Biomedicine, Neuroscience and Advanced Diagnostic Department, Unit of Ophthalmology, University of Palermo, Palermo, Italy, 10 Department of Surgical Sciences, University of Torino, Turin, Italy, 11 Eye Clinic, IRCCS MultiMedica, Milan, Italy, 12 Department Experimental Medicine, Tor Vergata University, Viale Oxford, Roma, Italy

¶ The details of EAGLE study principal investigators is provided in Acknowledgments.
* giovanni.staurenghi@unimi.it

## Abstract

### Objective

To assess the overall effectiveness of anti-vascular endothelial growth factor (VEGF) therapy in treatment-naïve patients with neovascular age-related macular degeneration (nAMD) in a clinical practice setting.

### Study design

EAGLE was a retrospective, 2-year, cohort observational, multicenter study conducted in Italy that analyzed secondary data of treatment-naïve patients with nAMD. The primary endpoint evaluated the mean annualized number of anti-VEGF injections at Years 1 and 2. The main secondary endpoints analyzed the mean change in visual acuity (VA) from baseline and variables associated with visual outcomes at Years 1 and 2.

### Results

Of the 752 patients enrolled, 745 (99.07%) received the first dose of anti-VEGF in 2016. Overall, 429 (57.05%) and 335 (44.5%) patients completed the 1- and 2-year follow-ups,

**Data Availability Statement:** All relevant data are within the manuscript and its Supporting Information files.

**Funding:** This study was funded by Novartis Farma S.p.A, Origgio, Italy. Elena Peruzzi (EP), Stefania Bassanini (SB) and Chiara Biancotto (CB) are employees of Novartis Farma S.p.A., Origgio, Italy. The funder provided support in the form of salaries for authors [CB,EP,SB]. The sponsor had a role in the study design, study conduction, data collection, data analysis, data interpretation and manuscript preparation. Additionally, Novartis Farma S.p.A was responsible for the conduct of the study and oversight of the collection and management of data. The specific roles of the authors employed by Novartis Farma are articulated in the 'author contributions' section.

**Competing interests:** Giovanni Staurenghi: Heidelberg Engineering1,2,3, Optos2, Ocular Instruments4, Optovue2, Quantel Medical2, Centervue1,2, Carl Zeiss Meditec2, Nidek2,3, Apellis1, Allergan1, Astellas1, Bayer1,3, Boheringer1, Topcon2, Genentech1, Iveric1, Novartis1,3, Roche1,3, Chengdu Kanghong Biotechnology Co1, Kyoto Drug Discovery & Development Co1. 1 consultant/advisor, 2 grant support, 3 lecture fee, 4 patents/royalty Francesco Bandello: Advisory for Allergan, Bayer, Boehringer-Ingelheim, Fidia Sooft, Hofmann La Roche, Novartis, NTC Pharma, Sifi, Thrombogenics and Zeiss Francesco Viola: Advisor for Allergan, Bayer, Novartis Farma S.p.A., and Roche. Monica Varano: Advisory board participation for Allergan, Bayer, Biogen, and Novartis Farma S.p.A. Giulia Barbati: Statistical consultant and received funding from Novartis. Elena Peruzzi, Stefania Bassanini and Chiara Biancotto: Employees of Novartis Farma S. p.A., Origgio, Italy. Vito Fenicia: Nothing to declare Claudio Furino: Advisory board participation for Allergan and Bayer Maria Vadalà: Advisor for Allergan Italia S.p.A., Bayer Italia S.p.A., and Novartis Farma S.p.A. Michele Reibaldi: Advisor for Allergan, Bayer, Novartis and SIFI. Stela Vujosevic: Advisory Board participation for Allergan, Apellis, Bayer, and Novartis. Federico Ricci: Advisor for Allergan, Bayer, Biogen, Genetech, MS&D, Novartis, Regeneron, and Roche. This aforementioned disclosures/ commercial affiliations of the authors does not alter our adherence to PLOS ONE policies on sharing data and materials.

respectively. At baseline, mean (standard deviation, SD) age was 75.6 (8.8) years and the mean (SD) VA was 53.43 (22.8) letters. The mean (SD) number of injections performed over the 2 years was 8.2 (4.1) resulting in a mean (SD) change in VA of 2.45 (19.36) ($P$ = 0.0005) letters at Year 1 and −1.34 (20.85) ($P$ = 0.3984) letters at Year 2. Linear regression models showed that age, baseline VA, number of injections, and early fluid resolution were the variables independently associated with visual outcomes at Years 1 and 2.

## Conclusions

The EAGLE study analyzed the routine clinical practice management of patients with nAMD in Italy. The study suggested that visual outcomes in clinical practice may be improved with earlier diagnosis, higher number of injections, and accurate fluid resolution targeting during treatment induction.

## Introduction

Age-related macular degeneration (AMD) is a progressive degenerative disease affecting the retina and is a leading cause of severe irreversible vision loss in the elderly, if left untreated. With its high prevalence and a progressively aging population, AMD is expected to affect 288 million by 2040 [1,2], leading to serious social consequences [3]. The neovascular form of AMD (nAMD/late-stage AMD) occurs mainly due to abnormally high expression of vascular endothelial growth factor (VEGF) [4–6] resulting in pathologic angiogenesis that determines the growth of blood vessels underneath the macula. The newly formed blood vessels are immature, and leak fluid (and sometimes blood) into the retina, disrupting its architecture, leading to progressive, severe, irreversible retinal damage [7].

Anti-VEGF therapies have revolutionized the treatment of nAMD [6,8,9]. Their efficacy in the maintenance of patient's visual acuity (VA), owing to their mode of action in keeping the macula dry by inhibiting the recurrence of fluid, has been demonstrated in many pivotal trials [6,10–13]. Early detection, diagnosis, prompt therapeutic intervention, and continuous follow-up to assess fluid accumulation and other activity signs are critical to prevent irreversible vision loss; nonetheless, these are difficult to achieve in clinical practice, thus creating a gap between clinical trial and real-world results [8].

In Italian routine clinical practice, there is a need for a descriptive analysis to assess the effectiveness of anti-VEGF treatment in the broad patient population and to identify the variables of different responses to these drugs. The **E**vidence of **A**nti-VE**G**F use in real **L**ife **E**xperience (EAGLE) study described here presents the current Italian routine clinical practice scenario and investigates the major factors associated with VA outcomes and nAMD management to leverage them for future perspectives in therapy.

## Materials and methods

### Study design

EAGLE was a retrospective, 2-year, cohort observational, multicenter study conducted at 27 clinical sites across Italy. The study enrollment period was from 1st January 2016 to 31st December 2016. Secondary data retrieved from hospital charts were analyzed and the main variables were collected at the index date (date of the first injection in treatment-naïve patients), and during the 2-year follow-up period, whose end was set at 31st December 2018.

The study protocol was reviewed and approved by Institutional Review Boards or Independent Ethics Committees. The complete list of investigational sites and related Ethics Committees are listed in S1 Table. All required local approvals from Ethics Committees were obtained before commencing data collection at each site.

### Key eligibility criteria

EAGLE enrolled treatment-naïve patients with a confirmed nAMD diagnosis, who started on-label anti-VEGF therapy between 1st January 2016 and 31st December 2016. Patients provided written and signed informed consent for study inclusion and reviewing of charts.

### Study objectives and endpoints

The primary objective was to evaluate anti-VEGF injections performed in clinical practice in patients with nAMD, treated for the first time with an anti-VEGF licensed for intraocular use, with respect to the mean (annualized) total number of anti-VEGF injections at Year 1, Year 2, and over 2 years.

The key secondary objectives were to (1) evaluate changes in VA from baseline at Years 1 and 2 in the treated eye, (2) evaluate factors associated with VA outcomes in the treated eye at Years 1 and 2 (age, gender, baseline VA, time from diagnosis to treatment, baseline type of macular neovascularization [MNV]; [14]), loading phase [LP], number of injections in the first year of treatment, and bilateral diagnosis), (3) evaluate factors associated with VA outcomes in the treated eye at Years 1 and 2 in the subgroup of patients who completed the LP, defined as patients receiving at least the first 3 injections in 90 days, and (4) estimate the median survival time of observation (overall exposure) from the index date to specific time points (6, 12, 18, and 24 months) stratified by baseline VA.

### Assessments

Effectiveness assessments included annualized number of anti-VEGF injections to evaluate mean values and absolute changes in VA (Early Treatment Diabetic Retinopathy Study [ETDRS]) at Years 1 and 2 after the start of anti-VEGF therapy in the treated eye (compared with baseline). The association of variables such as age, gender, baseline VA, time from diagnosis to treatment, number of injections in the first year of treatment, LP, bilateral diagnosis and baseline type of MNV lesion on VA outcomes of the treated eye were assessed by means of linear regression models at Years 1 and 2 in the whole population and in the subgroup of patients who completed the LP. LP patients were classified as 'wet' or 'dry' based on the presence or absence of fluid in the treated eye at the corresponding optical coherence tomography (OCT) evaluation and by investigator's judgment.

### Statistical analysis

Owing to the descriptive nature of the study, the statistical analyses associated with the primary endpoint (number of injections) and the secondary endpoint (change in VA) are descriptive; therefore, no formal statistical hypotheses have been stated. Sample size calculations were estimated using MedCalc Statistical Software version 18.5 (MedCalc Software bvba, Ostend, Belgium) and all statistical analyses were performed using software R, version 3.6.3. Sample size calculations referred to the desired precision for the main outcome estimate (i.e., the average number of injections per year). The main secondary endpoint (i.e., the estimated mean change in VA) was also taken into account. With respect to the primary outcome, it was calculated that 668 patients were required to estimate the mean number of injections per year with a 99%

probability of obtaining a confidence interval (CI) with a width of not more than 1, assuming a standard deviation (SD) of 5 for the mean number of injections. Regarding the main secondary endpoint, it was calculated that 668 patients were also required to estimate the mean change from a baseline score in VA based on letter count with a 99% probability of obtaining a CI with a width of not more than 2 letters, assuming a SD for the difference distribution of 10. Assuming that 10% of enrolled patients will have only one measure, at least 742 patients were expected to be enrolled. During the enrollment period, this size was respected considering a substantial loss because some patients refused to give their informed consent or due to difficulty in reaching them 3 years after the start of the therapy; thus, around 1300 patients' charts were needed to be screened.

The primary endpoint of the study was estimated in terms of mean (annualized) total number of injections calculated at the end of the first year (Month 12), the second year (i.e., between Months 13 and 24), and overall, at Year 2 (Month 24) and reported with the corresponding 95% CI. The following analysis populations/groups were considered: Overall Exposed (OE, defined as all enrolled patients who received at least one dose of anti-VEGF treatment during the enrollment period i.e., from 1st January to 31st December 2016) and Efficacy Analysis (EA, defined as all patients in the OE who had a baseline and at least one post-baseline VA assessment) sets categorized by i) type of MNV lesion at baseline, ii) baseline VA and visual impairment classes for the treated eye, iii) LP versus no loading phase (NLP) sets and iv) 'dry' versus 'wet' patients after the LP.

To evaluate factors associated with VA outcomes, linear regression models were estimated at 1 and 2 years in the First-year Completer Analysis set (1stCA_EA) and Second-year Completer Analysis set (2ndCA_EA) populations, using the following covariates: age, gender, baseline VA, time from diagnosis to treatment, baseline type of MNV lesion, bilateral diagnosis, number of injections in the first year of treatment, and LP completed. Retinal fluid at the end of LP was evaluated as a covariate (instead of LP completed) in similar regression models on VA outcomes at 1 and 2 years in the groups of wet and dry patients with an available VA assessment at 1 and 2 years, respectively.

The subset of independent variables associated with the mean change in VA was estimated using a multivariable analysis approach, starting from a full model and retaining variables with $P < 0.05$. As a supplementary analysis, to describe changes in VA during the follow-up using all available measurements per patient, a regression Linear Mixed Effects Model (LMM) with a random intercept for each patient was implemented using time as covariate (modeled as a restricted cubic spline).

## Results

### Patient disposition, demographics, and baseline ocular characteristics

**Patient disposition.** Of the 1336 patients screened for study enrollment, 752 were deemed eligible and signed informed consent, and were therefore included as the Enrolled Population (EP). Of these, 745 (99.07%) were in the OE set and 617 (82.05%) were included in the EA set. In the EA set, 429 patients had an available follow-up evaluation of VA at least 1 year after the first injection and were categorized as First-year Completer Analysis set (1stCA_EA), while 335 had an available follow-up evaluation of VA at least 2 years after the first injection (2ndCA_EA). Furthermore, 452 patients from the EA set completed the LP (LP population). Of the 366 patients in the LP who had an available OCT performed between the end of LP and the 4th injection, 191 were classified as 'wet' nAMD and 175 were classified as 'dry' nAMD as deemed by the investigator based on the presence or absence of fluid after the LP (**Fig 1**).

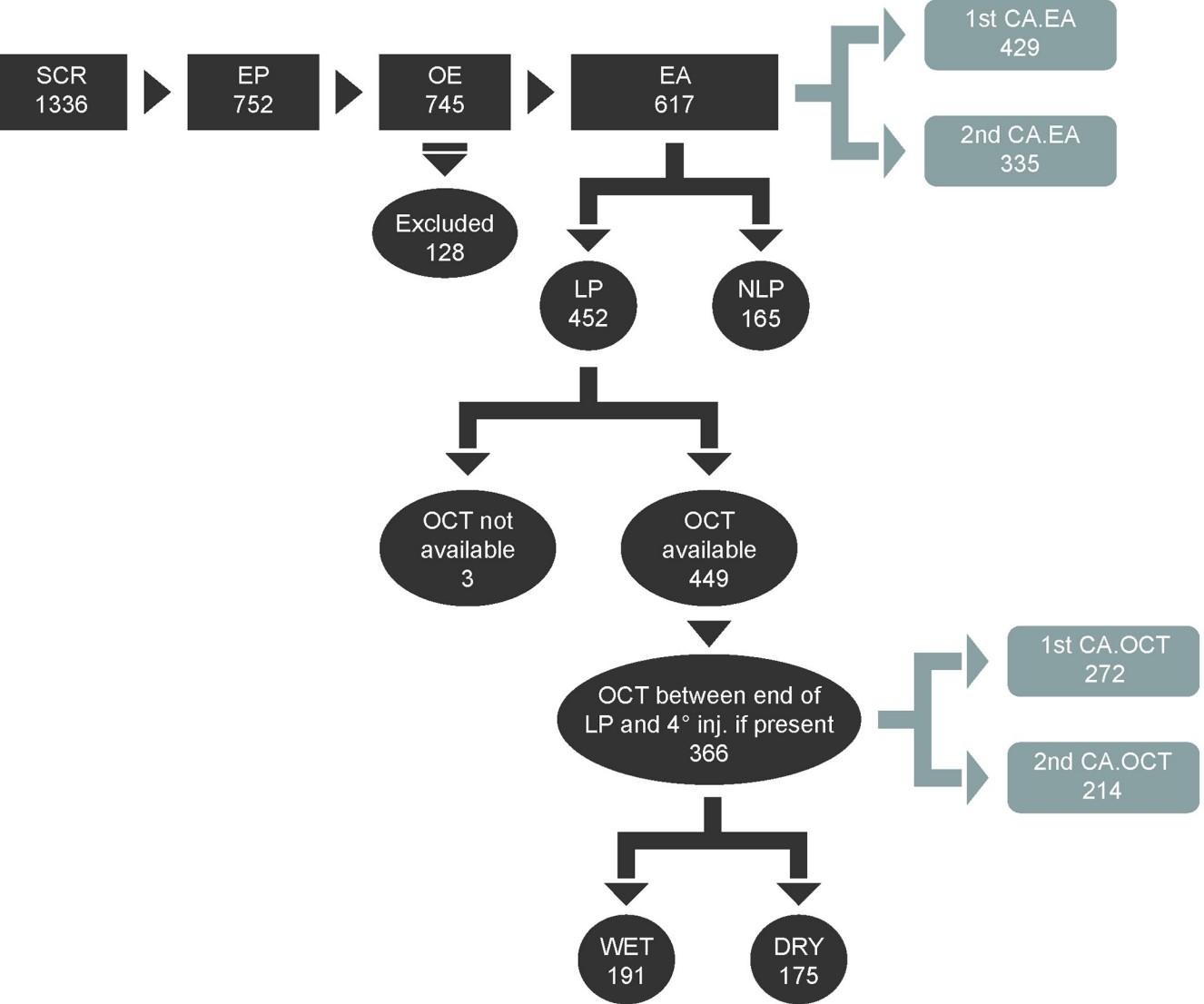

**Fig 1. Disposition of patients screened and enrolled in the study analysis.** Screened population (**SCR**): All patients who were screened, including those who did not give consent, but were contacted by the investigators; Enrolled population (**EP**): All screened patients who were eligible (i.e. fulfilled all inclusion and exclusion criteria) and who gave consent to participate in the study or were dead; Overall Exposed population (**OE**): All enrolled patients who received at least one dose of an anti-VEGF treatment during the enrollment period; Effectiveness Analysis set (**EA**): All patients in the OE who had a baseline and at least one post-baseline assessment of VA; **Excluded**: All patients included in the OE but not in the EA; Loading Phase population (**LP**): All patients in the EA who received at least 3 injections within 3 months (90 days) from the index date, with the date of the end of the LP for each patient being the date of the third injection in this time window; Not complete Loading Phase (**NLP**): All patients included in the EA but not in the LP; First-year Completer Analysis set (**1stCA _EA**): All patients in the EA with an available follow-up evaluation of VA at least 1 year after the first injection; Second-year Completer Analysis set (**2ndCA _EA**): All patients in the EA with an available follow-up evaluation of VA at least 2 years after the first injection; First-year OCT completer analysis set (**1stCA_OCT**): All patients in the LP with an available follow-up OCT assessment at least 1 year after the first injection; Second-year OCT completer analysis set (**2ndCA _OCT**): All patients in the LP with an available follow-up OCT assessment at least 2 years after the first injection. **Wet** and **Dry** classification (patients in the LP with an available OCT evaluation performed after the end of LP and before the date of the subsequent injection): A patient was classified as "dry" if at the corresponding OCT evaluation there was no presence of fluid in the treated eye based on investigator's judgment, while if at the corresponding OCT evaluation there was presence of fluid in the treated eye based on investigator's judgment, he/she was classified as "wet".

**Demographics and baseline ocular and disease characteristics.** As shown in **Table 1**, the EA population was nearly all Caucasian (612 [99.19%]) and more than half were female (EA: 335 [54.29%]). The mean (SD) age at the index date was 75.5 (8.7) years. The baseline mean VA,

**Table 1. Demographics and baseline ocular and disease characteristics.**

| Parameters | Effectiveness Analysis (EA) population (N = 617) |
|---|---|
| Mean (SD) age, years | 75.5 (8.7) |
| Gender, Female, n (%) | 335 (54.29) |
| Race, Caucasian, n (%) | 612 (99.19) |
| **Time from diagnosis to treatment (days)** | |
| n (%) | 584 (94.65) |
| Median | 16 |
| Q1, Q3 | 7; 33 |
| **VA, ETDRS letters** | |
| Mean (SD) | 53.43 (22.8) |
| Median (Q1; Q3) | 60 (35–70) |
| **MNV types (treated eye), n (%)** | |
| Classic (type II) | 167 (27.07) |
| Mixed | 52 (8.43) |
| ND | 109 (17.67) |
| Occult (type I) | 218 (35.33) |
| PCV | 27 (4.38) |
| RAP | 44 (7.13) |
| **CSRT (μm)** | |
| n (%) | 396 (64.18) |
| Mean (SD) | 395.7 (143.1) |
| Median (Q1;Q3) | 369.5 (300.0;463.0) |
| **OCT variables** | |
| n (%) | 481 (100) |
| Presence of fluid (investigators' judgment) | 436 (90.64) |
| Intra-retinal fluid | 289 (60.08) |
| Sub-retinal fluid | 314 (65.28) |
| RPE detachment | 285 (59.25) |
| Atrophy | 56 (11.64) |
| Fibrosis | 73 (15.18) |
| **Ocular disease history** | |
| n(%) | 247 (40.03) |
| Cataract | 38 (6.15) |
| Cataract surgery | 0 (0) |
| Vitrectomy | 0 (0) |
| RPE tear | 0 (0) |
| Other | 43 (6.96) |

OE population: All enrolled patients who had at least one anti-VEGF injection; EA set: All patients in the OE who had a baseline and at least one post-baseline assessment of VA.

CSRT, central sub-field retinal thickness; EA, effectiveness analysis; ETDRS, early treatment diabetic retinopathy study; LP, loading phase; MNV, macular neovascularization; N, total number of patients; n, number of patients; ND, not determined; OE, overall exposed; PCV; polypoidal choroidal vasculopathy; RAP, retinal angiomatous proliferation; RPE, retinal pigment epithelium; SD, standard deviation; VA, visual acuity; VEGF, vascular endothelial growth factor.

central subfield retinal thickness (CSRT), and presence of fluid (as per investigators' discretion) are also presented. Patients most frequently presented with type I (218 [35.33%]) and type II (167 [27.07%]) MNV lesions in the study eye (lesion type was classified using OCT or data entered

into case report forms [CRFs]) (**Table 1**). Baseline characteristics observed in the EA set were comparable with those of the OE (**S2 Table**). In the OE and EA populations, 167 and 135 patients, respectively, were observed to have developed bilateral nAMD during the observation period. No significant difference in the type of MNV lesions in the treated eye was observed between patients with unilateral or bilateral nAMD. Overall exposure in the EA population was stratified by baseline VA categories ($<58$, $\geq58$, and $<70$, $\geq70$ Early Treatment Diabetic Retinopathy Study [ETDRS] letters). A statistically significant difference ($P<0.02$) was noted in the estimated survival curve of observation time between patients with baseline VA $<58$ compared with those with baseline VA$\geq70$ ETDRS letters. Patients with better VA at baseline ($\geq70$ ETDRS) had a longer observation period compared with patients with worse baseline VA ($<58$ ETDRS) (**S1 Fig**).

## Annualized number of anti-VEGF injections

The mean (SD) number of anti-VEGF injections in the EA set was 5.6 ($\pm$ 2.5) at Year 1, 3.0 ($\pm$ 3.1) during Year 2 and 8.2 ($\pm$ 4.1) during the overall 2-year study period (**Fig 2**). The median (Q1;Q3) time from diagnosis to treatment was 16 (7;33) days in the EA set (**Table 1**). The number of injections and time from diagnosis were comparable between the EA and OE sets (**S2 Fig**; **S2 and S3 Tables**). Patients in the LP population set had a statistically significant higher mean (SD) numbers of injections compared with patients in the NLP in Years 1 and 2 and over 2 years (**S3A Fig**). Moreover, at the end of Year 1, the mean (SD) number of injections was significantly higher in LP patients who had 'wet' status after the LP compared with 'dry' patients (**S3B Fig**).

The time interval between consecutive injections during Year 1, Year 2, and over 2 years is shown in **Table 2**.

## Annualized number of injections based on baseline ocular characteristics

**Number of injections based on MNV lesion type at baseline.** A statistically significant difference in the mean total annualized number of anti-VEGF injections among MNV lesion

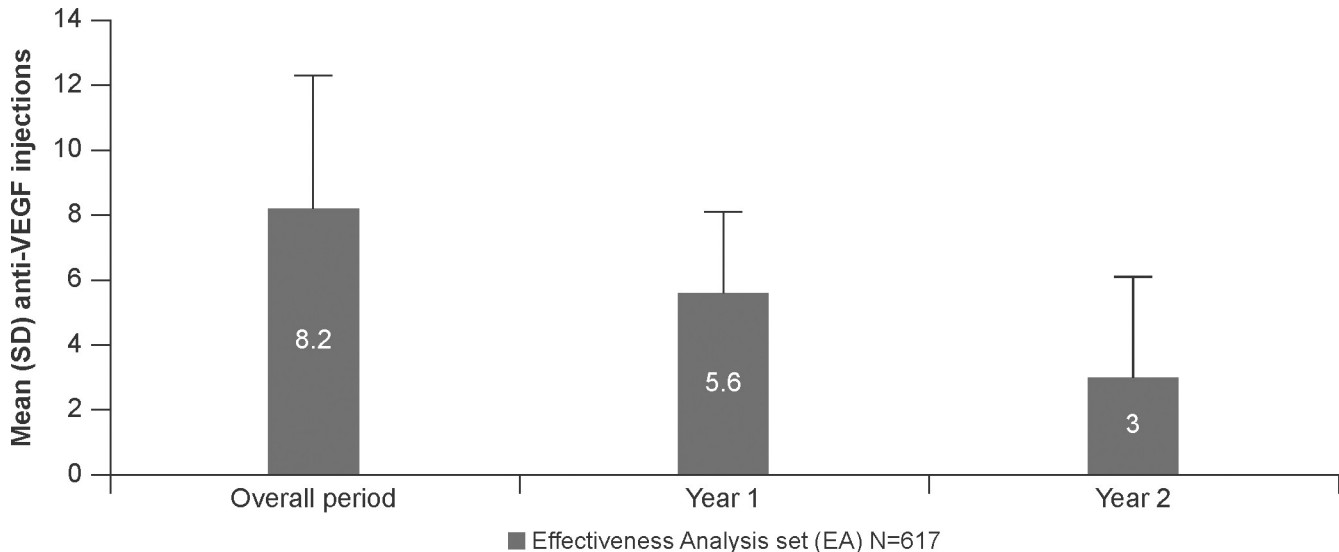

**Fig 2. Mean (SD) annualized anti-VEGF injections in EA study population.** OE population: All enrolled patients who had at least one anti-VEGF injection; EA set: All patients in the OE who had a baseline and at least one post-baseline assessment of VA. Mean (SD) number of anti-VEGF injections received by EA populations during Year 1 (until Month 12), Year 2 (Months 13–24) and overall period are presented. EA, effectiveness analysis; n, number of patients; OE, overall exposed; SD, standard deviation; VA, visual acuity; VEGF, vascular endothelial growth factor.

types was seen in Year 2 ($P = 0.0049$) and over 2 years ($P = 0.0255$) but not in Year 1. The mean total annualized number of injections appeared to be higher in patients with polypoidal choroidal vasculopathy and lower in those with classic lesions (S4A Fig).

**Number of injections based on baseline VA.** The annualized number of anti-VEGF injections was stratified by baseline VA categories ($<58$, $\geq58$ and $<70$, $\geq70$ and $<79$, and $\geq79$ ETDRS letters) as well as by visual impairment categories identified by The International Classification of Diseases 11 (2018) (see Supplementary data, S4C Fig). The mean (SD) total number of injections was statistically different among VA disjoint classes at Year 1 ($P = 0.0038$), between Years 1 and 2 ($P<0.0001$), and over 2 years ($P<0.0001$) in the EA population. Patients with better VA received a higher mean number of injections compared with those with lower baseline VA (S4B Fig).

## VA outcomes over time after anti-VEGF therapy

At Year 1, the mean (SD) VA increased by 2.45 ($\pm19.36$) letters compared with baseline; the change was statistically significant ($P = 0.0005$). Patients who were kept on treatment throughout the observation period maintained their VA with a mean (SD) VA change at Year 2 from baseline that was –1.34 ($\pm20.85$) letters with $P = 0.3984$ (S4 Table).

Furthermore, to describe changes in VA through follow-up using all available measurements per patient, a LMM regression model with a random intercept for each patient was implemented using time as a covariate (modelled as a restricted cubic spline). As shown in Fig 3, when using all available VA measures per patient, a significant positive time effect was detected followed by a decrease in VA after the first 200 days post-baseline VA assessment.

To evaluate factors associated with VA outcomes, linear regression models were estimated at Years 1 and 2 in the 1stCA_EA and 2ndCA_EA populations, using the following covariates: age, gender, baseline VA, time from diagnosis to treatment, baseline type of MNV lesion, number of injections during the first year, bilateral diagnosis, and loading phase completion. The subset of independent factors associated with VA outcomes at 1 and 2 years were age ($P = 0.0276$ [Year 1]; $P = 0.0043$ [Year 2]), baseline VA ($P<0.0001$ [Years 1 and 2]), and the number of injections during the first year ($P<0.0001$ [Year1]; $P = 0.0394$ [Year2]) (Table 3).

The persistence of retinal fluid at the end of the LP was also evaluated as a covariate in similar regression analysis on VA outcomes in the subgroup of LP patients with available VA assessments at 1 and 2 years. Results indicated that the presence of retinal fluid after the LP was significantly associated with VA outcomes: VA at Years 1 and 2 decreased on average by 4.7143 ($P = 0.0364$) and 5.0244 ($P = 0.0536$) letters, respectively, for patients with persistent fluid compared with patients with 'dry' retina (Table 4).

## Safety outcomes

No systemic safety data were collected owing to the retrospective nature of the study.

**Table 2. Time between consecutive injections.**

| Statistical parameters | EA population | | |
|---|---|---|---|
| | Year 1 (n = 617) | Year 2 (n = 524) | Overall (n = 617) |
| **n** | 597 | 310 | 602 |
| **Mean (SD) (days)** | 52 (23.5) | 87.5 (56.2) | 62.9 (50.8) |
| **95% CI** | 50.2–53.9 | 83–91.9 | 58.9–66.9 |

Interval in days were computed as: [date of j+1 injection]—[date of j injection] + 1.

CI, confidence interval; EA, effectiveness analysis; n, number of patients who had at least two injections; SD, standard deviation.

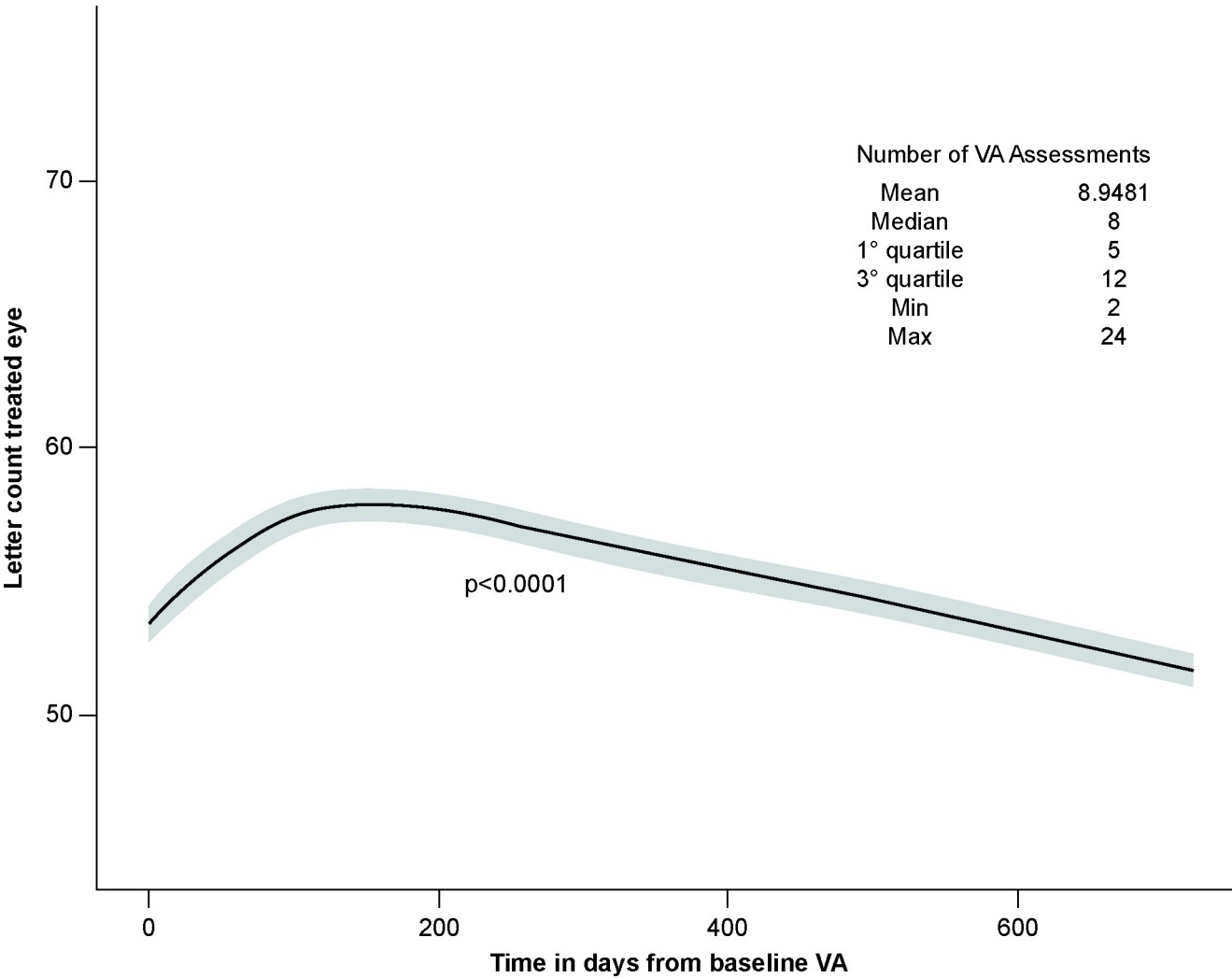

**Fig 3. Estimated letter count trend for the EA population during the study period.** Letter count trend in the treated eye during the study period is depicted as a function of time from baseline VA. The mean number of assessments per patient in the overall study period was 8.95. EA set: All patients in the OE who had a baseline and at least one post-baseline assessment of VA; EA, effectiveness analysis; OE, overall exposed; VA, visual acuity.

## Discussion

EAGLE was a 2-year, cohort observational, retrospective, multicenter study in Italy that evaluated the effectiveness of anti-VEGF therapy conducted on secondary data retrieved directly from hospital charts. It is also the first study in Italy to provide insights about the management of treatment-naïve patients with nAMD in clinical practice and to evaluate factors affecting responses to anti-VEGF treatment after 1 and 2 years.

The mean number of anti-VEGF injections in the EAGLE study at Year 1 was 5.6, an improvement on the previous routine clinical practice study, AURA (5.2 in 2 years for Italy) [15]. Despite this improvement, the mean number of injections clearly shows that patients in the EAGLE study were undertreated with respect to pivotal randomized controlled trials [10–12,16] and other routine clinical practice studies, particularly those adopting a proactive and customized treatment approach, with the goal of preventing disease recurrence [17,18].

The completion of a loading scheme was important for receiving anti-VEGF injections at a higher frequency during both the first and second year of follow-up, as well as overall (overall

**Table 3. Multivariable regression model results for ETDRS change after the 1st and 2nd year excluding patients with ND lesion type (EA set, 1stCA and 2ndCA).**

| Variable | ETDRS change after Year 1 | | | | ETDRS change after Year 2 | | | |
|---|---|---|---|---|---|---|---|---|
| | Estimate | SE | t value | Pr(>\|t\|) | Estimate | SE | t value | Pr(>\|t\|) |
| Intercept | 27.2041 | 9.4969 | 2.8645 | 0.0044 | 42.3832 | 11.4797 | 3.6920 | 0.0003 |
| **Age (1-year increase)** | **−0.2365** | **0.1069** | **−2.2120** | **0.0276** | **−0.3989** | **0.1384** | **−2.8817** | **0.0043** |
| Gender: male vs female | 1.3970 | 1.7916 | 0.7797 | 0.4361 | 0.3504 | 2.2320 | 0.1570 | 0.8754 |
| **Baseline VA (ETDRS)** | **0.6336** | **0.0436** | **14.5254** | **0.0000** | **0.6125** | **0.0540** | **11.3456** | **0.0000** |
| Time to treatment (1-day increase) | −0.0011 | 0.0112 | −0.0944 | 0.9248 | −0.0166 | 0.0269 | −0.6177 | 0.5373 |
| Lesion type: mixed vs classic | −2.5713 | 3.0990 | −0.8297 | 0.4073 | −0.5398 | 3.9204 | −0.1377 | 0.8906 |
| Lesion type: occult vs classic | −1.1198 | 2.0889 | −0.5361 | 0.5923 | 3.6140 | 2.6664 | 1.3554 | 0.1765 |
| Lesion type: PCV vs classic | 0.0608 | 4.2996 | 0.0142 | 0.9887 | 9.7109 | 5.0502 | 1.9229 | 0.0556 |
| Lesion type: RAP vs classic | −0.0890 | 3.3535 | −0.0265 | 0.9788 | −4.8802 | 4.0313 | −1.2106 | 0.2271 |
| **Number of injections in the first year (1-unit increase)** | **2.5666** | **0.5266** | **4.8744** | **0.0000** | **0.6873** | **0.3320** | **2.0704** | **0.0394** |
| Loading phase completed: no vs yes | 2.1128 | 2.1982 | 0.9612 | 0.3371 | 2.7693 | 2.7403 | 1.0106 | 0.3131 |
| Bilateral diagnosis: yes vs no | 0.6254 | 2.0628 | 0.3032 | 0.7619 | 3.6440 | 2.5898 | 1.4070 | 0.1606 |

*Adjusted R$^2$ for the model is 0.44, **Patients with complete data were 354 for Year 1.

*Adjusted R$^2$ for the model is 0.39, **Patients with complete data were 275 for Year 2.

IstCA, first-year completer analysis set; 2ndCA, second-year completer analysis set; ETDRS, early treatment diabetic retinopathy study ND, not determined; PCV, polypoidal choroidal vasculopathy; RAP, retinal angiomatous proliferation; SE, standard error; VA, visual acuity.

period: LP, 8.7; NLP, 6.7; $P<0.0001$) and data demonstrates that more injections correlates with better VA outcomes. Other than completion of the LP, a better VA at baseline was the factor associated with a higher number of injections. In particular, data suggests that patients with better baseline VA were able to follow more appropriate treatment in terms of number of injections and treatment persistence. The reasons for this could be two-fold; lower baseline VA could be associated with increased fibrosis and disorganization of neurosensory layers at

**Table 4. Multivariable regression model results for VA changes between baseline, 1st year and 2nd years excluding patients with ND lesion type (wet vs dry set, 1stCA, 2ndCA).**

| Variable | VA change after Year 1 | | | | VA change after Year 2 | | | |
|---|---|---|---|---|---|---|---|---|
| | Estimate | SE | t value | Pr (>\|t\|) | Estimate | SE | t value | Pr (>\|t\|) |
| Intercept | 36.6528 | 11.3307 | 3.2348 | 0.0014 | 45.6233 | 13.5096 | 3.3771 | 0.0009 |
| **Age (1 year increase)** | **−0.3347** | **0.1244** | **−2.6899** | **0.0077** | **−0.3982** | **0.1583** | **−2.5152** | **0.0128** |
| Gender: male vs female | 2.2303 | 2.1679 | 1.0288 | 0.3047 | 0.1000 | 2.5592 | 0.0391 | 0.9689 |
| **Baseline VA (ETDRS)** | **0.5946** | **0.0538** | **11.0539** | **0.0000** | **0.5724** | **0.0596** | **9.6021** | **0.0000** |
| Time to treatment (1 day increase) | 0.0003 | 0.0118 | 0.0288 | 0.9771 | 0.0188 | 0.0377 | 0.4970 | 0.6199 |
| Lesion type: Mixed vs classic | −3.1347 | 3.5904 | −0.8731 | 0.3836 | 0.9366 | 4.4350 | 0.2112 | 0.8330 |
| Lesion type: occult vs classic | 0.1421 | 2.5443 | 0.0558 | 0.9555 | 5.2651 | 3.0005 | 1.7547 | 0.0811 |
| Lesion type: PCV vs classic | 1.3318 | 5.6620 | 0.2352 | 0.8143 | 11.8228 | 6.0292 | 1.9609 | 0.0516 |
| Lesion type: RAP vs classic | 0.5021 | 3.7943 | 0.1323 | 0.8948 | −5.3562 | 4.2839 | −1.2503 | 0.2129 |
| **Number of injections in the first year (1-unit increase)** | **3.0740** | **0.6276** | **4.8978** | **0.0000** | **0.8718** | **0.3812** | **2.2872** | **0.0234** |
| **Presence of fluid end LP: yes vs no** | **−4.7143** | **2.2393** | **−2.1052** | **0.0364** | **−5.0244** | **2.5849** | **−1.9438** | **0.0536** |
| Bilateral diagnosis: yes vs no | 2.7481 | 2.4866 | 1.1051 | 0.2703 | 3.3126 | 2.8576 | 1.1592 | 0.2480 |

*Adjusted R$^2$ for the model is 0.45, **Patients with complete data were 225 for year 1.

*Adjusted R$^2$ for the model is 0.43, **Patients with complete data were 178 for year 2.

IstCA, first-year completer analysis set; 2ndCA, second-year completer analysis set; ETDRS, early treatment diabetic retinopathy study; LP, loading phase; PCV, polypoidal choroidal vasculopathy; RAP, retinal angiomatous proliferation; SE, standard error; VA, visual acuity.

baseline and thus, a more rapid evolution of scar. Furthermore, physician might be reluctant to treat patients with a lower chance of improving VA based on the risk-benefit ratio assessment and the way patients with better baseline VA might be more motivated to continue therapy [19]. Although the reasons that confer a lower baseline VA at diagnosis may be multiple (such as age, other ocular pathologies, presence of geographic atrophy), data suggest the need for earlier diagnosis, earlier treatment, and better awareness on disease chronicity [20], the improvement of which might contribute to reduction in the high number of patients lost to follow-up in the first year of therapy (~43% (n = 323)).

With respect to functional results, mean VA increased during Year 1 of treatment (2.45 letters; *P* = 0.0005) and was maintained in Year 2 (−1.34 letters; *P* = 0.398). In line with reports in other real-world evidence studies, the extent of VA gain and its maintenance over time is associated with injection frequency [21–23]; receiving <7 injections in the first year does not guarantee a significant gain in letters with current available therapies [15,24]. A possible explanation for such an observation in clinical practice is the application of a flexible *pro re nata* regimen of treatment based on disease reactivation. This therapeutic scheme does not allow disease activity to be promptly detected and treated because monthly monitoring is challenging in clinical practice, resulting in the recurrence of exudative changes which culminate in unsatisfactory clinical outcomes [25]. In contrast, patients receiving proactive regimens (such as treat-and-extend or fixed dosing) are more likely to receive an adequate number of injections allowing VA gains in the first year that are maintained during follow-up years [22]. Apart from number of injections, the literature indicates that older age and a higher proportion of follow-up visits with active MNV lesions are associated with poor VA outcomes in patients with nAMD [26].

To better understand the variables associated with VA outcomes in Italian clinical practice, a multiple regression analysis was conducted in this study. As mentioned previously, the number of injections given in Year 1 was one of the principal factors determining a better VA at Years 1 and 2. In addition, the analysis revealed that age and baseline VA are the variables that had the most impact on visual gains supporting available evidences. In EAGLE, the completion of the LP *per se* had no significant effect on VA gains at 1 and 2 years compared with NLP subgroup; bigger real-world studies, such as AURA [15] and LUMINOUS [24], reported a tendency but did not demonstrate a strong association. These data suggest that in routine clinical practice, patients might take longer than the mandated number of days to complete the LP (3 injections) which could possibly result in milder VA outcomes [27]. Remarkably, EAGLE data showed that ~52% (191 out of 366) of patients who were assessed for fluid status after the LP, have unresolved fluid (wet) indicating this as one of the hurdles for better visual outcomes in current clinical practice, of which inappropriate LP might be one of the potential contributors.

Multivariable analysis in patients who completed the LP at Years 1 and 2, revealed that the presence of retinal fluid at the end of LP correlated with worse VA outcomes at Years 1 and 2. This even more highlights the importance of achieving an early dry condition to maintain a better VA in the long term. This observation from a clinical practice setting further confirmed what has already been demonstrated in *post hoc* analyses of pivotal studies [28,29], where early fluid control determined successful management of patients with nAMD, thereby decreasing the burden associated with number of injections. Thus, achieving a 'dry' retina is deemed to be more important for achieving better or desired long-term VA outcomes.

The results of EAGLE are representative of patients enrolled in an observational real-life setting and may not necessarily apply to all patients with nAMD. This study was subject to the limitations inherent to medical chart reviews. Moreover, this observational, retrospective study presented some methodological limitations, such as different clinical centers using different OCT machines and different retreatment criteria, which were not standardized.

Estimations of VA measurements are possible owing to conversions (decimals to ETDRS). MNV lesion type was classified based on the data entered into CRFs from different centers or OCT assessments. Furthermore, the study might have unmeasured confounders (i.e., latent characteristics not taken into account in the regression models such as reactive or proactive individualized strategy).

To conclude, EAGLE represents a comprehensive database of nAMD patients, showing a reliable picture of management with approved anti-VEGF drugs in Italy from 2016–2018. As observed in other routine clinical practices, the present analysis showed a reduced number of injections and a great loss of patients at follow-up, suggesting that treatment centers could deliver a limited number of injections that are carried out for a limited period and varied from one patient to another. The data analyzed from this chart review confirmed that absolute VA gains were always higher in patients with better VA at baseline, strengthening the need of prompt treatment with anti-VEGF drugs following diagnosis of nAMD. Patients with lower baseline VA ($<58$ letters) are usually lost early to follow-up and received an average of one less injection in the first year and 2 fewer injections in the second year compared with the rest of the study population. Moreover, the study demonstrated that in the clinical practice setting, early fluid resolution is a critical factor for achieving a better functional outcome. This study identified that in Italy, like many other countries, there is a critical unmet medical need linked to under-treatment of patients. More effective therapies targeting retinal fluid along with individualized proactive treatment strategies and stricter follow-ups are needed to achieve appropriate patient treatment in terms of efficacy, despite the limited treatment capability of the Italian healthcare system.

## Supporting information

**S1 Fig. Overall exposure by VA at baseline (EA population).** EA set: All patients in the OE who had a baseline and at least one post-baseline assessment of VA. At Month 12, 94% of patients with baseline VA $\geq70$ ETDRS letters, 87% of patients with baseline VA $\geq58$ and $<70$ ETDRS and 79% of patients with baseline VA $<58$ were under observation. At Month 24, 61%, 50% and 41% of patients with baseline VA $\geq70$, $\geq58$ and $<70$ ETDRS and $<58$ ETDRS letters, respectively, were still under observation. EA, effectiveness analysis; ETDRS, early treatment diabetic retinopathy study; VA, visual acuity. *According to the application of Bonferroni correction due to multiple comparisons the threshold of significance was set at 0.02.*
(TIF)

**S2 Fig. Mean (SD) annualized anti-VEGF injections in OE study population.** OE population: All enrolled patients who had at least one anti-VEGF injection; Mean (SD) number of anti-VEGF injections received by OE populations during Year 1 (until Month 12), Year 2 (Months 13–24) and overall period are presented. n, number of patients; OE, overall exposed; SD, standard deviation.
(TIF)

**S3 Fig. Mean (SD) annualized number of injections based on loading phase and in patients' classified wet vs dry.** Mean (SD) number of injections in patients during Year 1, 2 and overall period are presented based on (A) those completing LP or NLP; and (B) in patients classified as wet or dry based on investigators discretion at the end of LP. LP, loading phase; NLP, no loading phase; SD, standard deviation.
(TIF)

**S4 Fig. Mean (SD) annualized anti-VEGF injections based on baseline ocular characteristics in the EA population.** Mean (SD) annualized anti-VEGF injections in EA population

during the Year 1, 2 and overall period were evaluated based on baseline ocular characteristics (A) type of lesion, (B) baseline VA, and (C) visual impairment in the treated eye. EA, effectiveness analysis; SD, standard deviation; VEGF, vascular endothelial growth factor.
(TIF)

**S1 Table. List of EAGLE study investigational sites and ethics committees.**
(DOCX)

**S2 Table. Demographics and baseline ocular and disease characteristics in OE population.**
(DOCX)

**S3 Table Median (annualized) anti-VEGF injections in OE and EA study population.**
(DOCX)

**S4 Table. Summary statistics of VA (ETDRS letters) and change from baseline in ETDRS at 1 year and 2 years (EA population) as per available measures.**
(DOCX)

**S5 Table. List of EAGLE study investigators and affiliation.**
(DOCX)

**S1 Appendix. Statistical analysis.**
(DOCX)

## Acknowledgments

The authors thank all the EAGLE study investigators: Cristiana Laculli, Nicolò Massimo, Marco Nardi, Paolo Lanzetta, Leonardo Mastropasqua, Gianni Virgili, Francesco Boscia, Rosalia Giustolisi, Michele Coppola, Carlo Cagini, Vincenzo Pucci, Luca Migliavacca, Antonio Laborante, Enrico Peiretti, Tommaso Micelli Ferrari, Cesare Mariotti, Giuseppe Romeo for their valuable contribution towards this study. The membership/affiliation of EAGLE study investigators is provided in **S5 Table**. The authors also thank Swapna Ganduri (Senior Scientific Writer I, Medical and Clinical Solutions, NBS CO**NEXT**S, Novartis Healthcare Pvt. Ltd., Hyderabad, India) for medical writing and editorial assistance towards the development of this article.

## Author Contributions

**Conceptualization:** Giovanni Staurenghi, Giulia Barbati, Elena Peruzzi, Stefania Bassanini, Chiara Biancotto, Federico Ricci.

**Data curation:** Giovanni Staurenghi, Giulia Barbati, Elena Peruzzi, Chiara Biancotto, Federico Ricci.

**Formal analysis:** Giovanni Staurenghi, Giulia Barbati, Elena Peruzzi, Stefania Bassanini, Chiara Biancotto, Federico Ricci.

**Investigation:** Giovanni Staurenghi, Francesco Bandello, Francesco Viola, Monica Varano, Vito Fenicia, Claudio Furino, Maria Vadalà, Michele Reibaldi, Stela Vujosevic, Federico Ricci.

**Methodology:** Giovanni Staurenghi, Giulia Barbati, Elena Peruzzi, Stefania Bassanini, Chiara Biancotto, Federico Ricci.

**Supervision:** Giovanni Staurenghi, Giulia Barbati, Elena Peruzzi, Chiara Biancotto, Federico Ricci.

**Writing – original draft:** Giovanni Staurenghi, Francesco Bandello, Francesco Viola, Monica Varano, Giulia Barbati, Elena Peruzzi, Stefania Bassanini, Chiara Biancotto, Vito Fenicia, Claudio Furino, Maria Vadalà, Michele Reibaldi, Stela Vujosevic, Federico Ricci.

**Writing – review & editing:** Giovanni Staurenghi, Francesco Bandello, Francesco Viola, Monica Varano, Giulia Barbati, Elena Peruzzi, Stefania Bassanini, Chiara Biancotto, Vito Fenicia, Claudio Furino, Maria Vadalà, Michele Reibaldi, Stela Vujosevic, Federico Ricci.

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
