## [Decision Letter · Decision Letter 0]

17 Jun 2021

PONE-D-21-16791

Effectiveness of anti-vascular endothelial growth factors in neovascular age-related macular degeneration and variables associated with visual acuity outcomes: Results from the EAGLE study

PLOS ONE

Dear Dr. Staurenghi,

Thank you for submitting your manuscript to PLOS ONE. After careful consideration, we feel that it has merit but does not fully meet PLOS ONE’s publication criteria as it currently stands. Therefore, we invite you to submit a revised version of the manuscript that addresses the points raised during the review process.

We look forward to receiving your revised manuscript.

Kind regards,

Vikas Khetan, MD

Academic Editor

PLOS ONE

Journal Requirements:

1. Please ensure that your manuscript meets PLOS ONE's style requirements, including those for file naming. The PLOS ONE style templates can be found athttps://journals.plos.org/plosone/s/file?id=wjVg/PLOSOne_formatting_sample_main_body.pdf and https://journals.plos.org/plosone/s/file?id=ba62/PLOSOne_formatting_sample_title_authors_affiliations.pdf

2.  Thank you for including your ethics statement: "The study protocol was reviewed and approved by Institutional Review Board or Independent Ethics Committee. The complete list of investigational sites and related Ethics Committees are listed in S1 Table. All required local approvals from Ethics Committees were obtained before commencing data collection at each site."

a) Please amend your current ethics statement to include the full name of the ethics committee/institutional review board(s) that approved your specific study, as we have noted that information was not mentioned S1.

3. One of the noted authors is a group or consortium (EAGLE study investigators). In addition to naming the author group, please list the individual authors and affiliations within this group in the acknowledgments section of your manuscript. Please also indicate clearly a lead author for this group along with a contact email address.

'I have read the journal's policy and the authors of this manuscript have the following competing interests:

Giovanni Staurenghi: Heidelberg Engineering1,2.3, Optos2, Ocular Instruments4, Optovue2, Quantel Medical 2, Centervue1,2, Carl Zeiss Meditec2,  Nidek2,3, Apellis1, Allergan1, Astellas1, Bayer1,3, Boheringer1, Topcon2, Genentech1, Iveric1, Novartis1,3, Roche1,3, Chengdu Kanghong Biotechnology Co1 Kyoto Drug Discovery & Development Co1  1 consultant/advisor, 2 grant support, 3 lecture fee, 4 patents/royalty

Francesco Bandello: Advisory for Allergan, Bayer, Boehringer-Ingelheim, Fidia Sooft, Hofmann La Roche, Novartis, NTC Pharma, Sifi, Thrombogenics and Zeiss

Francesco Viola: Advisor for Allergan, Bayer, Novartis Farma S.p.A., and Roche.

Monica Varano: Advisory board participation for Allergan, Bayer, Biogen, and Novartis Farma S.p.A.

Giulia Barbati: Statistical consultant and received funding from Novartis.

Elena Peruzzi, Stefania Bassanini and Chiara Biancotto: Employees of Novartis Farma S.p.A., Origgio, Italy.

Claudio Furino: Advisory board participation for Allergan and Bayer

Maria Vadalà: Advisor for Allergan Italia S.p.A., Bayer Italia S.p.A., and Novartis Farma S.p.A.

Michele Reibaldi: Advisor for Allergan, Bayer, Novartis and SIFI.

Stela Vujosevic: Advisory Board participation for Allergan, Apellis, Bayer, and Novartis.

Federico Ricci: Advisor for Allergan, Bayer, Biogen, Genetech, MS&D, Novartis, Regeneron, and Roche.'

We note that one or more of the authors are employed by a commercial company: Novartis Farma.

Additional Editor Comments (if provided):

Please revise the manuscript as per the requirements of the reviewers.

Reviewers' comments:

Reviewer's Responses to Questions

**Comments to the Author**

1. Is the manuscript technically sound, and do the data support the conclusions?

Reviewer #1: Yes

Reviewer #2: Yes

2. Has the statistical analysis been performed appropriately and rigorously? 

Reviewer #1: No

Reviewer #2: I Don't Know

3. Have the authors made all data underlying the findings in their manuscript fully available?

Reviewer #1: No

Reviewer #2: Yes

4. Is the manuscript presented in an intelligible fashion and written in standard English?

Reviewer #1: No

Reviewer #2: Yes

5. Review Comments to the Author

Reviewer #1: Dear author,

I had the opportunity to review this manuscript and I am glad to share my viewpoints.

The study is undoubtedly good and sound. However, I have major concerns, and I would like to share these with you.

1. What is the exact message that this study convey that has not been conveyed by other studies, so far, including the study by Holz et al in 2015? The reports suggesting the number of injections, the visual gains, and the anatomic results in eyes with nAMD treated with AntiVEGF injections have already shown the real world data. How is it different from the Italian data that you are trying to present?

2. There is mention of the number of loading injections in your study, which is lass than the average number of injections in other studies. Understandably, there will be differences in various countries, following different protocols, and even clinician based variations. However, studies comparing between loading dose and PRN dose have reported minimal differences in visual acuity gains and anatomical results. However, the difference erupts in the ability to detect the amount of fibrosis/ atrophy at the presentation, which, in retrospective studies is a major confounding factor.

3. Studies have already proven the fact that the better visual acuity at baseline is correlated with overall better final visual acuity.

4. P16, Line 326: This statement may be over-exaggerated here. Retrospective chart analysis may not be able to define the exact type of fluid which was noted, and may miss important aspects such as RPE atrophy underlying some intraretinal fluid spaces suggesting poor outcomes.

5. P17, Line 336: This statement may not hold true in real world, especially when recent studies have reported no difference between "some fluid" versus "no fluid". Again, this is very subjective, and differs between subRPE fluid, subretinal fluid above the lesion, or subretinal fluid adjacent to a high RPED.

6. I have major concerns regarding the clinical implications of this study in real world situation.

7. Methodology gets the reader confused. Maybe, you can depict this in a line diagram format when this is being revised further.

8. How many patients had significant cataract/ cataract surgery/ any intravitreal procedures/ diabetic retinopathy/ atrophy at presentation etc. All these are extremely important confounders to the results of this study.

9. How many patients had IRF without SRF? How many of them had RPE atrophy underlying the fluid?

10. Was FFA done in all the patients to qualify for classic/ occult varieties? Most of the classic patterns on FFA would account for type 2 CNV, but this cannot be 100% true. Also, was ICGA done in all the patients especially when diagnosing conditions like PCV and RAP lesions?

11. There are plenty of grammatical and editing errors which need to be rectified.

Thank youy

Best regards

Reviewer #2: I wanted to know if the presence of retinal fluid at the end of LP included both intraretinal and subretinal fluid? In the discussion it is mentioned as remnant retinal fluid at the end of LP as one of the possible factors for reduced vision gain at the end of the study period.

Tolerable Subretinal fluid at the end of LP in the T and E regimens have shown comparable VA outcomes: referring to the FLUID study.

6. PLOS authors have the option to publish the peer review history of their article (what does this mean?). If published, this will include your full peer review and any attached files.

Reviewer #1: No

Reviewer #2: **Yes: **Chetan Rao

---

## [Author Response · Author response to Decision Letter 0]

23 Jul 2021

We thank the reviewers for their critical review and insightful comments on the manuscript. Following are our point-by-point responses to the reviewers’ queries and any changes in the manuscript arising out of the potential revisions. 

Reviewer 1: 

1. What is the exact message that this study convey that has not been conveyed by other studies, so far, including the study by Holz et al in 2015? The reports suggesting the number of injections, the visual gains, and the anatomic results in eyes with nAMD treated with Anti-VEGF injections have already shown the real world data. How is it different from the Italian data that you are trying to present?

Response: We thank the reviewer for the critical review of the manuscript.

We agree that consistent with other real-world (RW) studies, EAGLE demonstrates the relative 

under-treatment of patients with neovascular age-related degeneration (nAMD) receiving lower numbers of anti-vascular endothelial growth factor (VEGF) injections [in the second year specifically]. 

EAGLE study provides important insights about the real-life management of treatment-naïve patients with nAMD in Italy, as well as the determinants of the response to treatment after 1 and 2 years.

Multivariable analysis in patients, who completed the loading phase (LP; three consecutive injections in 90 days) at Years 1 and 2, revealed that the presence of retinal fluid at the end of the LP correlated with worse visual acuity (VA) outcomes at Years 1 and 2. (lines 284-289; lines 356-363 in track change version of the manuscript). Thus, a dry condition (i.e. absence of retinal fluid in the treated eye based on investigator’s judgment) along with an early diagnosis were found to be important prognostic factors for achieving and maintaining a better VA and a dry condition in the long-term. 

To our knowledge, no previous study has investigated the association between visual outcomes and the presence/absence of retinal fluid after the LP based on the investigator’s opinion in a RW study.

Furthermore, compared with the AURA study, EAGLE made it possible to update the data on the current scenario following the introduction to the market of aflibercept and it characterizes the Italian situation in detail. In fact, the number of Italian patients included in AURA was insufficient to assess subgroup-significant results.

2. There is mention of the number of loading injections in your study, which is less than the average number of injections in other studies. Understandably, there will be differences in various countries, following different protocols, and even clinician based variations. However, studies comparing between loading dose and PRN dose have reported minimal differences in visual acuity gains and anatomical results. However, the difference erupts in the ability to detect the amount of fibrosis/ atrophy at the presentation, which, in retrospective studies is a major confounding factor.

Response: The EAGLE study used secondary data from the primary clinical records of treatment-naïve patients with nAMD. We agree with the reviewer on the limitations associated with such retrospective chart review studies. 

In clinical practice, the study highlights that achieving a dry retina after the LP is critical for better VA outcomes at 1 and 2 years, despite confounding factors (such as treatment regimens).

Moreover, the study was able to obtain optical coherane tomography ( OCT) variables at baseline, end of LP, and Years 1 and 2 from the clinical charts of these patients. Hence, we do have details of the number of patients who presented with fibrosis/atrophy at these time points. We have included the details of the number of patients with atrophy and fibrosis at baseline in revised Table 1 (Pg:9-10, track changes).

This is descriptive piece of data and it mentions in the Discussion section (lines 316-321; track changes version) that, “lower baseline VA could be associated with increased fibrosis and disorganization of neurosensory layers at baseline and thus, a more rapid evolution of scar,’ but such a claim needs to be warranted and validated by further research.

Although these aspects are intriguing, this primary manuscript will address the primary and main secondary protocol endpoints. The contribution of fibrosis/atrophy to VA outcomes might be considered for future post hoc analyses in a secondary manuscript.

Changes in the manuscript: Pg:9-10; Revised Table 1 (track changes)

3. Studies have already proven the fact that the better visual acuity at baseline is correlated with overall better final visual acuity

Response: We agree with the reviewer. The EAGLE study, like any other real-world observational study, further demonstrates that higher/better baseline VA correlates with overall better final VA outcomes. In the present study, we tried to evaluate the reasons associated with this.

In fact, mean annualized anti-VEGF injections at Years 1 and 2 assessed based on baseline ocular characteristics and number of injections based on baseline VA reiterated the fact that patients with better VA received higher mean number of injections compared with those with lower baseline VA (prior Fig S5B, now Fig S4B in revised manuscript). Moreover, as described in prior Fig S2 (now Fig S1 in revised manuscript), patients with better VA at baseline (�70 Early Treatment Diabetic Retinopathy Study [ETDRS] letters) had a longer observation period compared with those with worse baseline VA (�58 ETDRS). These aspects are elaborated on in the Discussion section (lines 313-321; track changes version).It was possible to correlate better baseline VA with an early dry condition, leading to better VA outcomes.

The EAGLE study data thus suggests the need for earlier diagnosis, earlier treatment, and better awareness on disease chronicity, the improvement of which might contribute to a reduction in the high number of patients lost to follow-up in the first year of therapy.

Strikingly, in the multivariate analysis, which considers all associated variables, baseline VA is one of the most important factors that should be considered to improve the unmet need in wAMD. Locally strengthening this data is very important because it highlights a strong unmet medical need that is underestimated compared with other chronic diseases for which greater prevention is made.

4. P16, Line 326: This statement may be over-exaggerated here. Retrospective chart analysis may not be able to define the exact type of fluid which was noted, and may miss important aspects such as RPE atrophy underlying some intraretinal fluid spaces suggesting poor outcomes.

Response: The identification of the presence of fluid in the patients was based on investigator’s judgment at the corresponding OCT evaluation. 

We understand the limitations associated with retrospective medical chart reviews and associated methodological limitations have been emphasized (lines 364-373; track changes version). As pointed by the reviewer, we have now revised the statement (track changes: line 354)

Changes made in the manuscript: Pg 18; revised line 354, in track changes

5. P17, Line 336: This statement may not hold true in real world, especially when recent studies have reported no difference between "some fluid" versus "no fluid". Again, this is very subjective, and differs between subRPE fluid, subretinal fluid above the lesion, or subretinal fluid adjacent to a high RPED.

Response: We thank the reviewer for pointing this out. 

The recent study referred by you is the FLUID study, a randomized clinical trial (RCT), with inclusion/exclusion criteria where patients in both the intensive (who had subretinal fluid [SRF] resolution) and relaxed (who had some SRF) treatment groups received numerically higher number of injections during Year 1, compared to the total number of injections received by EAGLE study participants in 2 years. 

RCT’s are more controlled in terms of the time taken by the patients to complete the loading dose versus the RWE study that is described here. These could have attributed to non-inferior results seen with the ‘some SRF’ group compared with the ‘no SRF’ group.

In FLUID, SRF tolerance during treatment proved to be non-inferior compared with intensive treatment only when using a treat and extend approach; In EAGLE, treatment strategies (and under-treatment) could widely differ among the involved centers and hence it was not possible to assess if residual fluid was due to investigators’ intention to tolerate SRF.

The relative under-treatment seen in Italian clinical practice reprises the need for achieving a dry retina early for better VA outcomes. 

6. I have major concerns regarding the clinical implications of this study in real world situation.

Response: The study, in addition to describing the updated scenario of clinical practice in Italy for the treatment of wAMD, identifies critical factors to be improved in the management of patients suffering of wAMD in Italy: the study emphasizes the need for early diagnosis and, importantly, demonstrates the need for early fluid resolution in achieving better functional outcomes in a clinical practice setting. 

Locally strengthening this piece of data is important because it highlights a strong unmet medical need that is underestimated compared with other chronic diseases for which greater prevention is made.

7.Methodology gets the reader confused. Maybe, you can depict this in a line diagram format when this is being revised further

Response: Please refer to previous Fig S1 (now Fig 1), a schematic describing the disposition of patients screened and enrolled for study analysis that clearly depicts the number of patients in the Efficacy analysis (EA) set who had LP vs NLP (no loading phase) and the subsequent OCT assessments available to categorize them as ‘Wet’ or ‘Dry’. The prior Fig S1 is included in the main figures section (new Figure 1).

8. How many patients had significant cataract/ cataract surgery/ any intravitreal procedures/ diabetic retinopathy/ atrophy at presentation etc. All these are extremely important confounders to the results of this study.

Response: The most frequently reported ocular conditions at baseline were prior/ongoing cataract or prior cataract surgery. Revised Table 1 includes ocular disease history.

In the EA population, 247 (40.03%) concomitant ocular conditions were reported, of which 81 were ongoing during anti-VEGF treatment; 38 (6.15%) of these conditions were cataract, while none reported prior cataract surgery, retinal pigment epithelium (RPE) tear, or vitrectomy. Other ocular conditions were reported in 43 (6.96%) cases.

Changes made in the manuscript: Pg:9-10; Revised Table 1 (track changes)

9. How many patients had IRF without SRF? How many of them had RPE atrophy underlying the fluid?

Response: Details on the number of patients with intraretinal fluid (IRF), SRF, RPE detachment, atrophy, and fibrosis are presented in revised Table 1. These data are descriptive as the association between atrophy and baseline fluid type was not defined as a study endpoint. We acknowledge the reviewer for this suggestion and this might be considered for future post hoc analysis.

10. Was FFA done in all the patients to qualify for classic/ occult varieties? Most of the classic patterns on FFA would account for type 2 CNV, but this cannot be 100% true. Also, was ICGA done in all the patients especially when diagnosing conditions like PCV and RAP lesions?

Response: Lesion type was classified using OCT or data entered into CRFs (described in lines 199-201, lines 369-370 track changes version).In fact, overall, during the 24-month follow-up, patients had an average of 8.8±4.9 non-injection visits, 5.8±4.6 OCT assessments, but <1 evaluation of Fluorescein angiography (FAG), Indocyanine green angiography (ICGA) and Fundus autofluorescence (FAF). 

11. There are plenty of grammatical and editing errors which need to be rectified.

Response: We have addressed this point and the revised manuscript is now copy edited for language. 

Changes made in the manuscript: Editorial changes for language throughout the manuscript (in track)

Reviewer 2:

1. I wanted to know if the presence of retinal fluid at the end of LP included both intraretinal and subretinal fluid? In the discussion it is mentioned as remnant retinal fluid at the end of LP as one of the possible factors for reduced vision gain at the end of the study period.

Tolerable Subretinal fluid at the end of LP in the T and E regimens have shown comparable VA outcomes: referring to the FLUID study.

Response: Presence of fluid at the end of the LP (based on investigators judgment) refers to both IRF and SRF.

In the EAGLE study, the multi-variable regression analysis carried out indicated that the persistence of retinal fluid at the end of the LP was significantly associated with VA outcomes (Table 4). VA at Years 1 and 2 decreased on average by 4.7143 (P=0.0364) and 5.0244 (P=0.0536) letters, respectively, for patients with persistent fluid compared with patients with ‘dry’ retina. In addition, the manuscript has references supporting the fact that early dryness is associated with better visual outcomes and less treatment burden.

We agree that FLUID study (interventional RCT) demonstrated non-inferiority in VA gains between patient groups with some SRF fluid vs those with none. We feel that any comparison between a controlled interventional trial and a RW study should be done with some caution due to differences in study design, eligibility criteria, follow-up management, and treatment schedules. Moreover, in FLUID, both groups received a higher number of injections compared with RW studies and in FLUID study re-treatment criteria were pre-defined in the protocol.

In a RWE study, patients are often lost to follow-up early and also may take longer times to complete the loading dose than the mandatory days. Such factors might contribute to suboptimal results in a RW setting. This has been elaborated in lines 342-359 (track changes version).

---

## [Decision Letter · Decision Letter 1]

9 Aug 2021

Effectiveness of anti-vascular endothelial growth factors in neovascular age-related macular degeneration and variables associated with visual acuity outcomes: Results from the EAGLE study

PONE-D-21-16791R1

Dear Dr. Staurenghi,

We’re pleased to inform you that your manuscript has been judged scientifically suitable for publication and will be formally accepted for publication once it meets all outstanding technical requirements.

Kind regards,

Vikas Khetan, MD

Academic Editor

PLOS ONE

Additional Editor Comments (optional):

Reviewers' comments:

Reviewer's Responses to Questions

**Comments to the Author**

1. If the authors have adequately addressed your comments raised in a previous round of review and you feel that this manuscript is now acceptable for publication, you may indicate that here to bypass the “Comments to the Author” section, enter your conflict of interest statement in the “Confidential to Editor” section, and submit your "Accept" recommendation.

Reviewer #1: All comments have been addressed

Reviewer #2: All comments have been addressed

2. Is the manuscript technically sound, and do the data support the conclusions?

Reviewer #1: Yes

Reviewer #2: Yes

3. Has the statistical analysis been performed appropriately and rigorously? 

Reviewer #1: Yes

Reviewer #2: Yes

4. Have the authors made all data underlying the findings in their manuscript fully available?

Reviewer #1: Yes

Reviewer #2: Yes

5. Is the manuscript presented in an intelligible fashion and written in standard English?

Reviewer #1: Yes

Reviewer #2: Yes

6. Review Comments to the Author

Reviewer #1: Thank you for revising the manuscript extensively. All the data that was missing in the first document is now available, and I am glad that the manuscript meets a publication criteria. These results are extremely important to asses the adequate treatment of AMD in real world, although geographical and interpersonal differences may exist. I have no further comments

Reviewer #2: (No Response)

7. PLOS authors have the option to publish the peer review history of their article (what does this mean?). If published, this will include your full peer review and any attached files.

Reviewer #1: No

Reviewer #2: **Yes: **Chetan Rao

---

## [Editor Report · Acceptance letter]

23 Aug 2021

PONE-D-21-16791R1 

Effectiveness of anti-vascular endothelial growth factors in neovascular age-related macular degeneration and variables associated with visual acuity outcomes: Results from the EAGLE study 

Dear Dr. Staurenghi:

I'm pleased to inform you that your manuscript has been deemed suitable for publication in PLOS ONE. Congratulations! Your manuscript is now with our production department. 

Kind regards, 

on behalf of

Dr. Vikas Khetan 

Academic Editor

PLOS ONE